# OpenReview forum: "VLMGuard: Defending VLMs against Malicious Prompts via Unlabeled Data"
_ICLR.cc/2025/Conference — ICLR 2025 Conference Withdrawn Submission_

### Official Review · Reviewer_M4sW · 2024-10-21

**Soundness:** 3
**Presentation:** 3
**Contribution:** 3
**Rating:** 6
**Confidence:** 4

**Summary:**

This paper presents a novel approach to defending Vision-Language Models (VLMs) against malicious prompts. The proposed method leverages the intrinsic capabilities of VLMs to assign pseudo-labels to unlabeled data, and subsequently trains a binary classifier to detect malicious prompts using this pseudo-labeled dataset. By doing so, the method significantly enhances the robustness of VLMs in detecting and mitigating malicious prompts.

**Strengths:**

Clarity and Simplicity of the Approach:
The overall idea and methodology presented in this paper are highly intuitive and easy to follow. The process of feeding inputs into the model to obtain embeddings, followed by performing SVD, and finally identifying outliers, is clear and logically structured. This clarity allows for smooth comprehension of the workflow, making the contributions more accessible to both researchers and practitioners.

Significance in Addressing a Critical Problem:
The paper addresses a critical issue in trustworthy AI: the detection and defense against malicious prompts in VLMs. The significance of this contribution cannot be overstated, as enhancing VLMs' robustness to malicious inputs is a crucial step toward ensuring reliable deployment of AI systems in real-world applications. The focus on using pseudo-labeled, unlabeled data is particularly valuable in reducing the dependency on manually labeled datasets, which is often a bottleneck in scaling robust AI solutions.

Substantial Performance Improvement:
The experimental results demonstrate considerable improvements over existing baselines, showcasing the practical impact of the proposed method. The enhancements in robustness against malicious prompts, as reflected by the increased AUROC scores in the experiments, highlight the effectiveness of the approach. This strong empirical performance further strengthens the paper's contribution.

**Weaknesses:**

Limited Novelty in Core Contribution:
The core contribution of this work—applying SVD to detect malicious prompts—while effective, does not appear to be particularly novel. SVD has been extensively used in anomaly detection tasks across various domains, and its direct application here may lack the originality expected in top-tier conference submissions. The paper could benefit from further emphasizing any unique insights or enhancements introduced in the specific context of VLMs and malicious prompt detection, beyond the straightforward use of SVD.

Unclear Necessity of Training the Protective Prompt Classifier:
A significant question arises regarding the necessity of the protective prompt classifier. Given that the proposed “maliciousness estimation in the latent subspace” seems capable of distinguishing malicious samples effectively, the added step of pseudo-labeling and training a binary classifier may seem redundant. The paper could have provided stronger justification for why the classifier is needed, beyond simply determining the decision threshold \tau. It would strengthen the contribution if the authors could clarify whether the classifier introduces additional benefits in terms of generalization or robustness that are not achieved by the latent subspace estimation alone. Without this clarification, the added complexity might appear unnecessary.


**Post-Rebuttal**:

The authors have not addressed my concerns regarding the necessity of training a classifier. Overall, the paper attempts to implement some simple functions using complex modules, lacking true innovation. I am inclined to rate it between 5 and 6.

**Questions:**

The biggest issue I see is related to Weakness 2.

Relation to Adversarial Images:The relationship between this work and adversarial images needs further clarification. While malicious prompts are discussed, there seems to be some overlap with the concept of adversarial examples, especially when dealing with adversarial attacks on the visual input. Could the authors elaborate on how their approach relates to or differs from traditional adversarial attack detection.

Applicability Beyond Multimodal Models:The method proposed in this paper does not seem inherently restricted to multimodal models. Would the authors agree that this approach could also be applied to unimodal models, such as those in CV or NLP? For instance, could this method be adapted for adversarial attack detection in CV models, where the "jailbreak" prompts are replaced by adversarial attacks?

---

> ### Author Response · Authors · 2024-11-23
> **Response to Reviewer M4sW**
>
> We thank you for recognizing our work as highly intuitive and for studying an important challenge. We appreciate the reviewer's comments and suggestions, which we address below:
>
>
>
> **A1. Emphasis on unique insights**
>
> Great point raised! While it is true that SVD has been used in anomaly detection tasks, VLMGuard’s application and enhancements in the context of VLMs and malicious prompt detection introduce several novel insights:
>
> - *Multimodal Context:* Unlike prior works that apply SVD to unimodal datasets (e.g., tabular or image data), VLMGuard is the first work to specifically targets the multimodal embeddings of vision-language models for malicious prompt detection. These embeddings encode complex interactions between vision and text inputs, requiring tailored approaches to effectively separate benign and malicious prompts. Malicious prompts in VLMs are inherently different from conventional anomalies. They exploit vulnerabilities in both textual and visual modalities, which makes their detection more challenging than uni-modal detection (see parapgraph 1 in introduction). VLMGuard’s SVD-based analysis identifies a subspace capturing these multimodal adversarial patterns, which is a novel application of SVD.
> - *Unsupervised Framework:* Unlike traditional anomaly detection tasks that often rely on labeled anomalies for validation, the core contribution of VLMGuard is that it provides an innovative approach that operates entirely on unlabeled data without costly human annotation on the maliciousness of the prompts. This necessitated the development of a novel approach to use SVD effectively in this unsupervised setting, including careful handling of mixed distributions of benign and malicious prompts.
> - *Detailed Method Design:* In addition to SVD, our learning framework includes the calculation of the maliciousness score, derived from projections onto the SVD-identified subspace for maliciousness estimation. Finally, the scores are used to train a binary safeguarding prompt classifier. These novel technical contributions along with the comprehensive experimental evaluations on flagship VLMs and representative multimodal threat models offer valuable insights to the LM safety research community.
>
> **A2. Clarification on the binary classifier**
>
> Another great point! We are happy to clarify.
>
> - Firstly, we would like to point out that our results in **Figure 4** have confirmed that training an additional prompt classifier can indeed benefit the detection performance compared to directly thresholding using the maliciousness scores calculated on the unlabeled data. This provides practical justification.
> - Secondly, we would like to highlight the technical benefits of the classifier training. Intuitively, the maliciousness score, defined as the norm of projections onto the principal subspace (Eq. 7), effectively captures the degree of alignment with the subspace identified by SVD. However, this method assumes a simple threshold-based separation between benign and malicious samples. When the distributions of benign and malicious samples overlap (as shown in Appendix B, Figure 6), a linear threshold cannot optimally separate the two classes. In contrast, the binary classifier starts with the noisy labels derived from the maliciousness score but has the flexibility to learn a **non-linear decision boundary that accounts for additional separability cues present in the full representation space**. For instance:
>     - It can exploit higher-order relationships or interactions between dimensions in the activation space that the score-based projection cannot capture.
>     - It can adjust for systematic noise or bias in the score-based annotations by finding a boundary that better fits the true underlying structure.
> - From a theoretical perspective, noisy labels can still lead to optimal classifiers under certain conditions [1], such as when the noise is unbiased or the classifier has sufficient capacity to model the true decision boundary. While the maliciousness score measures alignment with the top singular vectors, it inherently ignores information outside the identified subspace. The classifier, trained on the **full representation**, has access to the remaining latent dimensions, which may contain complementary information useful for distinguishing malicious prompts. Moreover, the training process of the binary classifier can be viewed as a denoising operation, where the model learns to generalize from the noisy score-based labels to better approximate the true underlying data distribution. This enables it to achieve performance beyond the initial noisy annotations.
>
> We believe further rigorous analysis by exploring  noisy label learning literature can be a promising future direction! Thank you pointing this out!

---

> > ### Author Response · Authors · 2024-11-23
> > **Response to Reviewer M4sW (II)**
> >
> > **A3. Clarification on adversarial examples**
> >
> > Thank you for the suggestion! Below, we discuss both the connections and key differences:
> >
> > **Connections.** Both malicious prompt detection and adversarial example detection aim to identify inputs crafted to exploit model vulnerabilities. In the current research landscape of multimodal malicious prompt attack and defense, most of the threat models utilize the adversarial examples that are generated by PGD for attacking the vision branch, which is similar to the generation strategy (i.e., PGD) of traditional adversarial examples with a classification model.
> >
> > **Difference.** Firstly, traditional adversarial attack detection typically focuses on visually perturbed inputs designed to fool unimodal **discriminative** classifiers. Techniques often rely on detecting subtle pixel-level perturbations or anomalies in activation patterns that can lead to incorrect classification outputs. Our work addresses the broader scope of malicious prompts targeting multimodal **generative** VLMs that can trigger unintended generations. This includes textual prompts (e.g., adversarial instructions) and their interactions with visual adversarial examples. These attacks exploit cross-modal vulnerabilities, which require different detection strategies. Moreover, as far as we know, our approach is the first to harness unlabeled user prompts for help malicious prompt detection, which is  underexplored in the traditional adversarial attack detection literature.
> >
> > **A4. Applicability on adversarial attack detection**
> >
> > We agree with the reviewer that the proposed approach might be useful for adversarial attack detection in CV models. In fact, there are already several existing papers that tried PCA for defending traditional adversarial examples [2-4] (although they did not explore using the unlabeled frameworks as we did). We believe this could be an interesting extension to work on in the future!
> >
> >
> > [1] Natarajan et al., Learning with Noisy Labels, NIPS 2013.
> >
> > [2] Gupte et al., PCA as a defense against some adversaries
> >
> > [3] Hendrycks et al., Visible progress on adversarial images and a new saliency map, 2016
> >
> > [4] Jere et al., Principal component properties of adversarial samples, 2019

---

> > > ### Author Response · Authors · 2024-11-25
> > > **any comments on our rebuttal?**
> > >
> > > Dear reviewer M4sW,
> > >
> > > We wanted to touch base with you for the author-reviewer discussion. We trust you've had the opportunity to review our rebuttal, and we're eager to address any questions or comments you may have.
> > >
> > > Thank you once again for your time and dedication to this review process. We look forward to your response and to furthering the dialogue on our manuscript.
> > >
> > > Thanks,
> > >
> > > Authors

---

> > > > ### Comment · Reviewer_M4sW · 2024-11-28
> > > >
> > > > Thank you for the author's response. I have one point that I don't fully understand. Is the maliciousness estimation score you refer to in the original paper a scalar? Could you provide some examples to explain the complete training process of the binary classifier, including data labeling?

---

> > > > > ### Author Response · Authors · 2024-11-28
> > > > > **Response to Reviewer M4sW**
> > > > >
> > > > > Thank you for the question! Yes, the maliciousness estimation score is a scalar. We are happy to clarify the training process:
> > > > > - Given an unlabeled user prompt dataset with $n$ data points, we firstly extract their embedding with the VLM, and calculate the maliciousness estimation score which has the dimension of [$n$,1].
> > > > > - Then, we have a threshold $T$ (which is a hyperparameter determined on the validation set, as explained in the last paragraph of Section 4.1) to determine the proxy labels of the unlabeled user prompt dataset, i.e., if the score is bigger than the threshold, the prompt is labeled as malicious and vice versa.
> > > > > - After that, we train the safeguarding prompt classifier based on the embeddings of the user prompts and their proxy labels.
> > > > > - During inference, we can directly use the trained prompt classifier for malicious prompt detection.
> > > > >
> > > > > We hope this can answer your question. Please let use know if there is anything you want to discuss further.

---

> ### Comment · Reviewer_M4sW · 2024-11-30
>
> The author's response is consistent with my understanding.
> So, I have a significant **doubt**:
> First, the "maliciousness estimation score" is estimated, and then different positive or negative labels are assigned to the samples based on a threshold. A binary classifier is trained using these labeled data, and this classifier is then used to label unseen data. Since this is the case, why not directly label the unseen data based on the maliciousness estimation score? In other words, the maliciousness estimation score already serves as the ground-truth label for the binary classifier.

---

> > ### Author Response · Authors · 2024-11-30
> > **Response to Reviewer M4sW**
> >
> > Thank you for the response! To answer your question, we would like to remind the reviewer that the maliciousness estimation score **cannot** be directly used as the ground truth labels for the binary classifer because they are not probabilistic values between 0 and 1 (see Eq. 6 and figure 6)
> >
> > However, we appreciate the suggestion and proceed to normalize the scores via dividing every score by their maximal value and use as the soft labels for training the binary classifier. The results on detecting adversarial meta-instructions on LLaVA are shown as follows. We will be sure to include the discussion in the revised version.
> > |     Variants   | AUROC |
> > | ------ | ----- |
> > | soft labels |84.61|
> > | Ours |**92.87**|

---

### Official Review · Reviewer_9GQL · 2024-10-27

**Soundness:** 3
**Presentation:** 3
**Contribution:** 3
**Rating:** 5
**Confidence:** 4

**Summary:**

Vision-language models (VLMs) are crucial for understanding visual and textual information but are vulnerable to adversarial inputs, which can compromise their reliability in applications. Detecting these malicious prompts is essential to maintain trust in VLM outputs. A significant challenge is the limited availability of labeled benign and malicious data for developing effective classifiers.

To tackle this, the paper introduces VLMGUARD, a novel framework that utilizes unlabeled user prompts from real-world deployments for malicious prompt detection. These prompts contain both benign and malicious information. The approach includes an automated maliciousness estimation score to differentiate between benign and malicious samples within this unlabeled dataset, allowing for the training of a binary prompt classifier without requiring additional human annotations.

Extensive experiments demonstrate that VLMGUARD achieves superior detection performance, significantly surpassing existing state-of-the-art methods, thus offering a flexible and practical solution for real-world applications.

**Strengths:**

- This paper explores the defense in VLM malicious generations, giving a good reference to the research on this aspect.

- The proposed method VLMGUARD is simple but effective to achieve the defense, and the good performance obtained by the experiments strongly supports this point.

- The ablation study is organized well to clearly demonstrate the whole proposed method. And it makes the paper easy to follow.

**Weaknesses:**

- I am curious about why the binary classifier outperforms the direct use of the maliciousness score for detection, as illustrated in Fig. 4. The training dataset is based on an unlabeled dataset that has been annotated with maliciousness scores. Consequently, the accuracy of the binary classifier relies on the quality of these annotations, which in turn depends on the effectiveness of the maliciousness score detection. This raises the question: **is the upper bound of the binary classifier's performance essentially limited by the accuracy of the maliciousness score detection?** However, the results in Fig. 4 contrasts this view.

- I noticed that the values for $\pi$ are chosen from the set {0.001, 0.005, 0.01, 0.05, 0.1} as stated in Section 4.1. Is it possible for the proportion of malicious prompts to be higher, perhaps 0.5 or more? I am interested in understanding whether the proposed defense mechanism remains effective when faced with such a higher rate of malicious prompts.

**Questions:**

Listed in the weakness of the paper.

Score can be improved if concerns listed above are resolved.

---

> ### Author Response · Authors · 2024-11-23
> **Response to Reviewer 9GQL**
>
> We are deeply encouraged that you recognize our method to be simple and effective and give a good reference to the research on this aspect. We appreciate the reviewer's comments and suggestions, which we address below:
>
>
>
>
> **A1. Clarification on performance of the binary classifier**
>
> You raise a great point! The discrepancy between the binary classifier’s performance and the direct use of the maliciousness score stems from *the classifier’s ability to leverage additional representation information* and *refine decision boundaries beyond the initial score thresholding*. Based on this, we provide our detailed reasonings below:
>
> - Intuitively, the maliciousness score, defined as the norm of projections onto the principal subspace (Eq. 7), effectively captures the degree of alignment with the subspace identified by SVD. However, this method assumes a simple threshold-based separation between benign and malicious samples. When the distributions of benign and malicious samples overlap (as shown in Appendix B, Figure 6), a linear threshold cannot optimally separate the two classes. In contrast, the binary classifier starts with the noisy labels derived from the maliciousness score but has the flexibility to learn a **non-linear decision boundary that accounts for additional separability cues present in the full representation space**. For instance:
>     - It can exploit higher-order relationships or interactions between dimensions in the activation space that the score-based projection cannot capture.
>     - It can adjust for systematic noise or bias in the score-based annotations by finding a boundary that better fits the true underlying structure.
> - From a theoretical perspective, noisy labels can still lead to optimal classifiers under certain conditions [1], such as when the noise is unbiased or the classifier has sufficient capacity to model the true decision boundary. While the maliciousness score measures alignment with the top singular vectors, it inherently ignores information outside the identified subspace. The classifier, trained on the **full representation**, has access to the remaining latent dimensions, which may contain complementary information useful for distinguishing malicious prompts. Moreover, the training process of the binary classifier can be viewed as a denoising operation, where the model learns to generalize from the noisy score-based labels to better approximate the true underlying data distribution. This enables it to achieve performance beyond the initial noisy annotations.
>
>
> Based on these reasonings, we believe that the upper bound of the binary classifier's performance is not necessarily limited by the accuracy of the maliciousness score detection. However, we agree with the reviewer that they should be related under certain conditions. Further rigorous analysis by exploring  noisy label learning literature can be a promising future direction!
>
> **A2. Experimenting with higher maliciousness ratio**
>
> Absolutely! We intend to choose a small maliciousness ratio in order to simulate the real-world settings where the majority of input prompts are benign while only a small fraction of them are malicious. As suggested, we follow the same experimental setting as in Figure 3 (b), provide the experimental results when $\pi=0.3, 0.5, 0.7, 0.9$, and report the detection performance as follows. The results achieve near optimal performance and plateau when $\pi$ becomes larger. This precisely highlights that our approach only requires a small number of malicious samples for training a perfect binary malicious prompt classifier, which aligns with the practical application scenarios.
>
>
> |     Ratio   | AUROC |
> | ------ | ----- |
> | 0.01 |92.87|
> | 0.05|95.44|
> |0.1 | 98.95|
> |0.3 | 98.63|
> | 0.5 |98.72 |
> |0.7| 98.96 |
> |0.9|98.68 |
>
>
> [1] Natarajan et al., Learning with Noisy Labels, NIPS 2013.

---

> > ### Author Response · Authors · 2024-11-25
> > **any comments on our rebuttal?**
> >
> > Dear reviewer 9GQL,
> >
> > We wanted to touch base with you for the author-reviewer discussion. We trust you've had the opportunity to review our rebuttal, and we're eager to address any questions or comments you may have.
> >
> > Thank you once again for your time and dedication to this review process. We look forward to your response and to furthering the dialogue on our manuscript.
> >
> > Thanks,
> >
> > Authors

---

### Official Review · Reviewer_FGDR · 2024-11-02

**Soundness:** 2
**Presentation:** 2
**Contribution:** 2
**Rating:** 3
**Confidence:** 4

**Summary:**

This paper introduces a defensive framework called VLMGuard, designed to safeguard Vision-Language Models (VLMs) from malicious user inputs. VLMGuard approaches malicious prompt detection as a binary classification task, operating within the VLM's latent space. It identifies prompts that fall into a subspace defined by the latent vectors of known malicious prompts, effectively flagging them as toxic or jailbreaking attempts.

**Strengths:**

- This paper focuses on a critical safety issue: the misuse of VLMs through malicious or adversarial user inputs. This topic is increasingly important due to the growing popularity and widespread deployment of VLMs.

- VLMGuard introduces an interesting approach by utilizing unlabeled user inputs to enhance the detection of malicious content. This method presents a promising and effective solution to the problem.

**Weaknesses:**

1. **The motivation behind VLMGuard is unclear.** While it is purportedly designed for VLMs, the integration of VLM concepts into the method is not evident. VLMGuard appears to function as a general binary classifier using extracted latent features applicable to any deep neural network. The lack of a clear rationale and organized presentation diminishes the method's potential significance.

2. **The presentation is wordy and lacks informativeness.** The introduction fails to provide an overarching view of the method. Additionally, Figure 1 lacks annotations necessary for reader comprehension.

3. **There is insufficient discussion of novelty and technical contributions.** Although the authors highlight that VLMGuard requires no labeled data, it seems to rely on latent vectors of known malicious prompts. Furthermore, as indicated by Eqs 5, 6, and Figure 2, the solution resembles an SVM approach [a1]. The authors should clarify the novelty and contributions compared to existing methods.

[a1] M. A. Hearst, S. T. Dumais, E. Osuna, J. Platt, and B. Scholkopf, "Support vector machines," IEEE Intelligent Systems, vol. 13, no. 4, pp. 18-28, July-Aug. 1998, doi: 10.1109/5254.708428.

4. **There is a lack of comparison with closely related baselines.** The authors should compare their method with state-of-the-art content moderation solutions, such as Aegis [a2], LlamaGuard [a3], LlamaGuard2 [a3], LlamaGuard3 [a4], and OpenAI-Moderation [a5].


[a2] https://arxiv.org/abs/2404.05993
[a3] https://arxiv.org/abs/2312.06674
[a4] https://arxiv.org/abs/2407.21783
[a5] https://arxiv.org/pdf/2208.03274

5. **(minor) Figure 1 is misleading.** While the paper does not evaluate any of OpenAI's VLMs, it uses the OpenAI logo, potentially confusing readers. This should be corrected to accurately reflect the VLMs used in the evaluation.

**Questions:**

Please refer to the weaknesses section.

---

> ### Author Response · Authors · 2024-11-23
> **Response to Reviewer FGDR**
>
> We are glad to see that the reviewer recognized our work solve an increasingly important safety issue and the approach to be effective and interesting. We thank the reviewer for the thorough comments and suggestions. We are happy to clarify as follows:
>
>
> **A1. Clarification on the motivation**
>
> Thank you for pointing this out! We are happy to clarify.
>
> **Motivation.** Firstly, as illustrated in the **first two paragraphs of introduction** of our paper, the motivation for VLMGuard arises from the unique vulnerabilities of VLMs, which operate on multimodal inputs (text and vision) and integrate cross-modal information. Compared to unimodal models, VLMs face *greater* challenges in detecting malicious prompts because such prompts can exploit vulnerabilities in either modality or their interactions. Specifically, malicious prompts in VLMs often manifest as subtle adversarial perturbations to visual components or as carefully crafted textual instructions that disrupt the model's alignment. VLMGuard is designed to tackle these challenges by leveraging the distinctive properties of VLM embeddings, which encode *multimodal contextual information* to detect such malicious inputs effectively.
>
> **VLM concepts in VLMGuard.** The key component of VLMGuard is its ability to analyze the latent subspace in the VLM's embedding space. The embeddings reflect **multimodal alignment**, which can capture the nuanced interactions between vision and text inputs. *Unlike generic deep neural networks, VLMs integrate information across visual and textual modalities*.  As such, applying VLMGuard to a non-VLM model would require a fundamentally different interpretation of the latent subspace. VLMGuard operates directly on these cross-modal representations, and utilize their specific structure to identify anomalies indicative of malicious intent. During rebuttal, we empirically verify that using the multimodal embeddings shows better malicious prompt detection performance compared to uni-modal representations within the same model (comparison in the below table on detecting adversarial meta-instruction for LLaVA).
>
> |     Representations   | AUROC |
> | ------ | ----- |
> |Visual (uni-modal) |83.29|
> |Textual (uni-modal)| 52.19|
> |Cross-modal (ours) | **92.87**|
>
> Finally, our maliciousness estimation score leverages the multimodal alignment property to identify patterns specific to multimodal malicious prompts, such as adversarial meta-instructions and jailbreak attacks. In fact, our paper is the first work to study detecting different kinds of malicious prompts designed for the multimodal domain on prominent VLMs (e.g., LLaVA and Phi-3). These experiments demonstrate VLMGuard’s effectiveness in addressing VLM-specific threats, which we believe can reinforce its design focus and make our method a significant contribution to the research community.
>
> **VLMGuard is not just a general binary classifier.** While VLMGuard includes a binary classification step, its core contribution lies in leveraging the unique properties of the *unlabeled data* and their *VLM embeddings* to estimate maliciousness. The subspace decomposition step is tailored to capture the high-dimensional relationships and multimodal alignment inherent to VLMs, which makes it less directly applicable to generic neural networks. In contrast, general deep neural networks typically lack the structured multimodal embeddings present in VLMs. Therefore, applying VLMGuard to a non-VLM model would require a fundamentally different interpretation of the latent subspace.
>
> Thank you again for bringing this up!
>
> **A2. Presentation issues**
>
> Thank you for the question! We would like to refer the reviewer to **paragraphs 3 and 4 in the introduction** for the description of our proposed approach. In paragraph 3, we introduce the unlabeled modeling framework as an important problem setup. Subseqently, we focus on explaining the key components of the proposed learning method about how to utilize the unlabeled data for malicious prompt detection. We believe that is helpful for readers to understand our method.
>
> For Figure 1, we have included the annotations of the key concepts in our learning framework in the picture and then described the algorithm details in the caption.
>
> If you have any specific details that you want us to clarify and revise for the figure and introduction, please do not hesitate to leave your comment!

---

> > ### Author Response · Authors · 2024-11-23
> > **Response to Reviewer FGDR (II)**
> >
> > **A3. More discussions on the novelty and technical contributions**
> >
> > Thank you for the suggestion! We are happy to add more discussions on this.
> >
> > **Reliance on latent vectors of known malicious prompts.** We would like to highlight the requirement of the malicious prompts during training in VLMGuard is *firstly **unlabeled** and secondly **can be different from the test-time malicious data*** (Definition 3.1 and Section 4.1). This could significantly improve the flexiblity and practicality of our approach, because the malicious prompts in the unlabeled data can be easily collectible and do not have to be similar to the test-time malicious prompts that one want to detect. In Figure 3 (a), we show that VLMGuard can even generalize across different threat models. For instance, the model trained with the unlabeled adversarial meta-instructions is able to detect the multimodal jailbreak prompts in the test time, which demonstrates the weak assumptions required on the unlabeled data.
> >
> >
> > **Novelty.** VLMGuard introduces several innovations that distinguish it from existing approaches, including SVM-like methods, as follows:
> >
> > - *Unlabeled Data Utilization*: VLMGuard operates by entirely on unlabeled data and using a novel maliciousness estimation score based on SVD, without requiring labeled malicious prompts. This is a key difference from supervised methods like SVMs. In fact, Eqs.5 and 6 are illustrations of the SVD procedure rather than SVMs.
> > - *Subspace-Based Analysis*: Unlike SVMs, which rely on labeled data to optimize hyperplanes, VLMGuard identifies a small subspace where malicious patterns concentrate using unsupervised decomposition of multimodal embeddings. This focuses on separating malicious and benign prompts geometrically within the representation space.
> > - *Multimodal Focus*: VLMGuard leverages the unique properties of VLM embeddings, specifically their cross-modal alignment. This makes it uniquely suited for VLM-specific threats such as multimodal adversarial attacks, unlike generic binary classifiers or SVMs.
> >
> >
> > **A4. Comparison with baselines**
> >
> > Thank you for the suggested baselines! After taking a careful look, we found that Aegis, LlamaGuard, and LlamaGuard2 cannot support multimodal inputs. We therefore compare VLMGuard with "Llama-Guard-3-11B-Vision" and the OpenAI-Moderation tool "omni-moderation-latest", where the results on detecting jailbreak prompts are shown in the following table. We observed that the OpenAI-Moderation tool can achieve a pretty close performance to VLMGuard but still cannot outperform VLMGuard because our approach is able to utilize the additional unlabeled data for training compared to post-hoc inference for these moderation tools, where the accurate separation between the unlabeled benign and jailbreak prompts provides useful signal for test-time jailbreak detection. The malicious prompts are synthesized using Phi-3 model.
> >
> >
> >
> > |     Model   | AUROC |
> > | ------ | ----- |
> > |LlamaGuard3-Vision |89.19|
> > |OpenAI-Moderation | 93.71|
> > | VLMGuard (ours) | **95.74**|
> >
> > **A5. Issues on Figure 1**
> >
> > Certainly! We have updated  Figure 1 and removed the OpenAI logo.

---

> > > ### Author Response · Authors · 2024-11-25
> > > **any comments on our rebuttal?**
> > >
> > > Dear reviewer FGDR,
> > >
> > > We wanted to touch base with you for the author-reviewer discussion. We trust you've had the opportunity to review our rebuttal, and we're eager to address any questions or comments you may have.
> > >
> > > Thank you once again for your time and dedication to this review process. We look forward to your response and to furthering the dialogue on our manuscript.
> > >
> > > Thanks,
> > >
> > > Authors

---

### Official Review · Reviewer_S7im · 2024-11-03

**Soundness:** 3
**Presentation:** 3
**Contribution:** 3
**Rating:** 6
**Confidence:** 3

**Summary:**

This paper addresses a challenge in VLM security in VLM security - detecting malicious prompts without requiring labeled data. The key innovation lies in analyzing unlabeled user data through subspace analysis of VLM representations. While the approach shows promising results, several fundamental questions about its theoretical foundation and practical applicability need to be addressed.

**Strengths:**

- New Problem Definition:
  - The paper presents a practical solution to reduce dependency on labeled data, which is particularly valuable because manually labeling malicious prompts is time-consuming and expensive.

- Technical Approach:
  - The proposed maliciousness scoring mechanism uses VLM's internal representations, which is computationally efficient as it requires only a single forward pass.
  - The scoring function $\kappa_i = \frac{1}{k} \sum \left( \lambda_j \cdot \langle f_i, v_j \rangle^2 \right)$  combines information from multiple principal directions, making it more robust than single-direction approaches.
  - The framework can be easily integrated into existing VLM systems since it doesn't require architectural modifications.

**Weaknesses:**

- The authors offer some geometric intuition and empirical validation, but the theoretical foundation could be clearer in a few areas:
  - The choice of SVD subspace analysis, while effective empirically, lacks a solid theoretical basis to confirm its effectiveness in detecting malicious patterns.
  - The current geometric explanation would be stronger with a formal analysis showing why this property holds across different types of attacks.


- The authors use the common approach of last-token embeddings, but there are still a few questions:
  - How does information from earlier tokens affect detection?
  - What properties of the last token make it particularly suitable?
  - How robust is this choice across different prompt structures?

**Questions:**

- Given your geometric intuition about malicious samples "malicious samples may occupy a small subspace within the activation space":
  - Could you provide formal conditions under which this property is guaranteed?
  - How does this property relate to the VLM's training objectives?

---

> ### Author Response · Authors · 2024-11-23
> **Response to Reviewer S7im**
>
> We thank the reviewer for the comments and suggestions. We are encouraged that you recognize our problem definition to be new and practical and with an efficient and robust technical approach. We address your questions below:
>
> **A1. Clarification on the theoretical foundation**
>
> Thank you for the suggestion! While the focus of the submission is not on theory, we do think it is helpful to derive a rigorous understanding of our approach. We provide our reasonings as follows:
>
> - Firstly, we believe that the theoretical basis for using the SVD subspace analysis stems from foundational properties of singular value decomposition (SVD) as a dimensionality reduction method. Specifically, SVD identifies dominant modes of variation in data [1], which can encapsulate structural features of embeddings. In our case, we hypothesize and empirically demonstrate that malicious prompts often induce unique patterns in the representation space, which are effectively captured as primary modes of variation by SVD. This separability is supported by the distribution analysis in **Figure 6** (**Appendix B**), where malicious prompts exhibit distinct projection magnitudes.
>
>   To strengthen the theoretical grounding, we believe the exploration on **spectral analysis** [2] of the covariance matrix on the unlabeled data can be a promising next step. Specifically, malicious samples can be modeled as perturbations of the benign samples. Therefore, we could leverage the perturbation analysis theory to quantify the difference in eigenvalues between the top-$k$ components and the rest to assess the contribution of malicious prompts. A formal treatment of this hypothesis is a natural extension of our work.
>  - In terms of the generalization across different attacks, spectral analysis could be a useful theoretical analysis tool as well. For example, a simple starting point for analysis is to model the different attacks as a distribution drawn from a proxy distribution, such as Gaussians. Based on this proxy modeling, we could subsequently analyze the spectral gap between the benign and malicious samples for measuring their distinguishability across different attacks, which could be calculated on real-world datasets for empirical verification.
>
> **A2. Discussions on the last-token embeddings**
>
> Thank you for the suggestion! Below, we address each point in detail:
>
> - **How does information from earlier tokens affect detection?** In our approach, the last-token embedding is chosen because it aggregates contextual information from all preceding tokens during the forward pass through the transformer layers. This is a common practice in representation-based analysis [3,4], where the representation of the last token is designed to reflect the cumulative context of the input sequence, including earlier tokens. Empirically, we found that embeddings derived from earlier tokens (e.g., 1/3, 2/3 position of the input prompt) often capture incomplete contextual signals (see table below), leading to less effective separation of malicious and benign prompts. Here we report the performance on detecting adversarial meta-instruction using LLaVA.
>
> |     Token position   | AUROC |
> | ------ | ----- |
> |1/3 |80.29|
> |2/3 | 87.63|
> |End (ours) | **92.87**|
>
> - **What properties of the last token make it particularly suitable?** The last-token embedding encapsulates the model's "final understanding" of the input, influenced by all previous tokens, including cross-modal interactions in vision-language models. It often represents a culmination of the attention and transformation processes applied throughout the model's depth. Additionally, malicious prompts, especially jailbreaking prompts, often inject adversarial patterns towards the end of the input (which corresponds to the textual encoding part). The last token is thus more likely to reflect the intent and context of the complete prompt.
> - **How robust is this choice across different prompt structures?** We follow the recommended prompt format for the LLaVA (https://huggingface.co/llava-hf/llava-v1.6-vicuna-7b-hf) and Phi-3 models (https://huggingface.co/microsoft/Phi-3-vision-128k-instruct) in our experiments. In the table below, we use the LLaVA prompt format for Phi-3 and Phi-3 prompt format for LLaVA, and report the performance on detecting adversarial meta-instruction. We observe that prompt structure can indeed affect the detection performance, and the recommended prompt format for a certain VLM in their huggingface repository is better for malicious prompt detection.
>
>
> |     Variants   | AUROC |
> | ------ | ----- |
> | LLaVA (Phi-3 prompt) |89.01|
> |LLaVA (LLaVA prompt) | **92.87**|
> |Phi-3 (LLaVA format) | 87.96|
> |Phi-3 (Phi-3 format) | **92.11**|

---

> ### Author Response · Authors · 2024-11-23
> **Response to Reviewer S7im (II)**
>
> **A3. Discussions on the geometric intuition**
>
> You raise a great point! To the best of our knowledge, the rigorous analysis of our geometric intuition, i.e., how does the LLM representations encode knowledge is still an open-ended research question in LLM Interpretability research domain, and leads to multiple recent papers working on this [5-7]. That being said, we are more than happy to provide our own understandings on your questions!
>
> Firstly, we hypothesize the intuition that *"malicious samples may occupy a small subspace"* is based on the assumption that malicious prompts represent rare and distinct patterns compared to benign prompts in the training data. This might be theoretically grounded under the condition of the low mixing ratio ($\pi$). In practice, the language model (LM) is trained with much more benign texts over malicious texts to perform general NLP tasks. Therefore, in addition to encoding the discriminative information for benign/malicious separation in its first several principal component of the representation space, LM representation subsumes more information in order to perform the tasks like text continuation, instruction following and reasoning. Such information can include the sentiment, grammar, lexical semantics, etc.
>
>
> **Relation to the VLM's training objectives.** Since VLMs are trained to align representations of text and image inputs, often through contrastive objectives or cross-modal attention. These objectives encourage embeddings of benign prompts to occupy a smooth and coherent manifold in the representation space. Malicious prompts, designed to exploit weaknesses in these alignments, introduce outlier patterns that deviate from this manifold, naturally forming a separate subspace.
>
> During training, VLMs are predominantly exposed to benign samples. This leads to a strong alignment of benign prompts with the main directions of variance in the representation space, while malicious prompts, being unseen or adversarially crafted, disrupt this alignment and are projected into distinct regions.
>
> Moreover, malicious prompts often exploit gradient-sensitive directions in the model’s loss landscape (e.g., through adversarial perturbations). These sensitive directions tend to correspond to sparse, high-variance subspaces in the activation space, which our SVD-based method effectively identifies.
>
>
>
> [1] https://en.wikipedia.org/wiki/Principal_component_analysis
>
> [2] Einsiedler et al., Functional Analysis, Spectral Theory, and Applications
>
> [3] Li et al., Inference-Time Intervention: Eliciting Truthful Answers from a Language Model, NeurIPS 2023
>
> [4] Zou et al., Representation Engineering: A Top-Down Approach to AI Transparency, 2023
>
> [5] Zheng et al., On Prompt-Driven Safeguarding for Large Language Models, ICML 2024
>
> [6] Friedman et al., Interpretability Illusions in the Generalization of Simplified Models, ICML 2024
>
> [7] Marks et al., The Geometry of Truth: Emergent Linear Structure in Large Language Model Representations of True/False Datasets, COLM 2024

---

> > ### Author Response · Authors · 2024-11-25
> > **any comments on our rebuttal?**
> >
> > Dear reviewer S7im,
> >
> > We wanted to touch base with you for the author-reviewer discussion. We trust you've had the opportunity to review our rebuttal, and we're eager to address any questions or comments you may have.
> >
> > Thank you once again for your time and dedication to this review process. We look forward to your response and to furthering the dialogue on our manuscript.
> >
> > Thanks,
> >
> > Authors

---

> ### Comment · Reviewer_S7im · 2024-11-30
>
> Thank you for your response and the detailed explanation of the geometric intuition behind your SVD-based method.
>
> However, I'm curious to know more about how your method handles attacks that exploit interactions between modalities, such as those explored in "Dual-Key Multimodal Backdoors" (Walmer et al., CVPR 2022). These attacks rely on subtle, coordinated triggers across modalities, and the malicious behavior only emerges through specific cross-modal interactions.
>
> It would be helpful to understand how the SVD approach captures these more complex cross-modal relationships.  For instance, could you elaborate on whether the singular vectors are sufficient to effectively model such dependencies?  Perhaps a deeper analysis of the singular vectors corresponding to the smaller singular values might reveal subtle patterns indicative of cross-modal attacks.
>
> Alternatively, could you provide some insights based on your current experimental setup on how your method might perform against these types of attacks? Even if explicit tests against coordinated cross-modal attacks are not feasible at this stage, any  reasoning or analysis based on your existing findings would be valuable.
>
>
> [1] Walmer, Matthew et al. “Dual-Key Multimodal Backdoors for Visual Question Answering.” 2022 IEEE/CVF Conference on Computer Vision and Pattern Recognition (CVPR) (2021): 15354-15364.

---

> > ### Author Response · Authors · 2024-11-30
> > **Response to Reviewer S7im**
> >
> > Thank you for the suggestion the related attack paper! We do think it is a valuable paper to consider for the multimodal models.  Here is our thought on how our approach can defend against the attack:
> >
> > - Firstly, similar to the current experimental setup in VLMGuard, we can leverage the posioned VQA dataset (as shown in Figure 2 of the mentioned paper) as the unlabeled dataset.
> > - Then, one can extract the embeddings (or internal activations) of the posioned VQA dataset to perform SVD and calculate the maliciousness estimation score, train the binary detector and then use it for the test-time defense.
> > - We would like to additionally highlight one interesting aspect in the experiment: since the mentioned dual-key attack is trained to only trigger malicious outputs when both triggers are activated (which is different from multimodal jailbreak attack), it might be useful to examine the different combinations of four types of data in the unlabeled mixture, i.e., no trigger is activated, only visual trigger is activated, only textual trigger is activated and both triggers are activated. There could lead to many potential novel methods on how to design a better unlabeled dataset for defending against the dual-key attack in the multimodal domain.
> >
> > We hope this can answer your question. Please let use know if there is anything you want to discuss further.

---

### Author Response · Authors · 2024-11-23
**General Response**

We thank all the reviewers for their time and valuable comments. We are encouraged to see that all reviewers find our approach **interesting, simple, novel, effective, robust, efficient**, and **has a new, significant and important problem setup** (S7im, FGDR, 9GQL, M4sW), and our results **superior and strong** (9GQL, M4sW). Reviewers also recognize our paper presentation to be **clear, well organized, and easy to follow** (9GQL, M4sW).

As recognized by multiple reviewers, the significance of our work can be summarized as follows:


- Our work offers a new algorithmic framework that leverages the unlabeled data in the models' deployment environments to help malicious prompt detection, which is an important research question.
- The framework is based on the factorization of the VLM representations, where the maliciousness of the unlabeled data is inferred and subsequently, a final safeguarding prompt classifier is learned. The approach is simple, clear, and effective.
- We provide supportive experiments to show the effectiveness of our approach, precisely highlighting how the proposed framework works in practice. Sufficient ablations are provided to help readers understand the method.

We respond to each reviewer's comments in detail below. We are happy to revise the manuscript according to the reviewers' suggestions, and we believe this will make our paper stronger.

---

### Author Response · Authors · 2024-11-30
**general response to all reviewers**

Dear Reviewers,

We hope this message finds you well. As the closing date for the review process is approaching, we wanted to follow up to ensure that all of your concerns regarding our paper have been addressed. If there are any remaining questions or issues you would like us to clarify, we would be more than happy to discuss them further.

If all your concerns have been resolved, we kindly ask if you could consider revisiting your scores and updating them to reflect the clarified contributions of our work. Your constructive feedback has been invaluable, and we sincerely appreciate the time and effort you have devoted to reviewing our paper.

Thank you once again for your thoughtful evaluations, and we look forward to hearing from you.

Best,

Authors

---

### Note · Authors · 2024-12-08

**Comment:**

Dear Reviewers,

We highly appreciate your constructive comments and questions. As we find many of the key questions deserve careful scrutiny and require in-depth experiments, the authors have decided to withdraw the submission for this round.

Thank you once again for your time and effort in going over our manuscript, not to mention the insightful comments to help us improve the paper.

Best regards,

Authors.

**Withdrawal Confirmation:**

I have read and agree with the venue's withdrawal policy on behalf of myself and my co-authors.